# Risk Awareness as a Key Determinant of Early Vaccine Uptake in the Mpox Vaccination Campaign in an Italian Region: A Cross-Sectional Analysis

**DOI:** 10.3390/vaccines11121761

**Published:** 2023-11-27

**Authors:** Giulia Del Duca, Alessandro Tavelli, Ilaria Mastrorosa, Camilla Aguglia, Simone Lanini, Anna Clelia Brita, Roberta Gagliardini, Serena Vita, Alessandra Vergori, Jessica Paulicelli, Giorgia Natalini, Angela D’Urso, Pierluca Piselli, Paola Gallì, Vanessa Mondillo, Claudio Mastroianni, Enrica Tamburrini, Loredana Sarmati, Christof Stingone, Miriam Lichtner, Emanuele Nicastri, Massimo Farinella, Filippo Leserri, Andrea Siddu, Fabrizio Maggi, Antonella d’Arminio Monforte, Francesco Vairo, Alessandra Barca, Francesco Vaia, Enrico Girardi, Valentina Mazzotta, Andrea Antinori

**Affiliations:** 1Clinical Infectious Diseases Department, National Institute for Infectious Diseases Lazzaro Spallanzani IRCCS, 00149 Rome, Italy; giulia.delduca@inmi.it (G.D.D.); camilla.aguglia@gmail.com (C.A.); simone.lanini@inmi.it (S.L.); annaclelia.brita@inmi.it (A.C.B.); roberta.gagliardini@inmi.it (R.G.); serena.vita@inmi.it (S.V.); alessandra.vergori@inmi.it (A.V.); jessica.paulicelli@inmi.it (J.P.); giorgia.natalini@inmi.it (G.N.); angela.durso@inmi.it (A.D.); emanuele.nicastri@inmi.it (E.N.); valentina.mazzotta@inmi.it (V.M.); andrea.antinori@inmi.it (A.A.); 2Icona Foundation, 20142 Milan, Italy; alessandro.tavelli@gmail.com (A.T.); antonella.darminio@unimi.it (A.d.M.); 3Department of System Medicine, University of Rome Tor Vergata, 00133 Rome, Italy; sarmati@med.uniroma2.it; 4Department of Epidemiology, National Institute for Infectious Diseases Lazzaro Spallanzani IRCCS, 00149 Rome, Italy; pierluca.piselli@inmi.it (P.P.); francesco.vairo@inmi.it (F.V.); 5Health Direction, National Institute for Infectious Diseases Lazzaro Spallanzani IRCCS, 00149 Rome, Italy; paola.galli@inmi.it (P.G.); vanessa.mondillo@inmi.it (V.M.); 6Department of Public Health and Infectious Diseases, Sapienza University, AOU Policlinico Umberto 1, 00161 Rome, Italy; claudio.mastroianni@uniroma1.it; 7Fondazione Policlinico A. Gemelli, Catholic University of the Sacred Heart, 00168 Rome, Italy; enrica.tamburrini@unicatt.it; 8STI/HIV Unit, San Gallicano IRCCS Dermatological Institute, 00144 Rome, Italy; christof.stingone@ifo.it; 9Neuroscience Mental Health and Sense Organs Department, Sapienza University of Rome, Polo Pontino, 04100 Latina, Italy; miriam.lichtner@uniroma1.it; 10Circolo di Cultura Omosessuale Mario Mieli, 00146 Rome, Italy; m.farinella@mariomieli.org; 11Plus Roma, 00182 Rome, Italy; checkpoint@plusroma.it; 12General Directorate of Prevention, Ministry of Health, 00197 Rome, Italy; a.siddu@sanita.it (A.S.); f.vaia@sanita.it (F.V.); 13Laboratory of Virology, National Institute for Infectious Diseases Lazzaro Spallanzani IRCCS, 00149 Rome, Italy; fabrizio.maggi@inmi.it; 14Unit of Health Promotion and Prevention, Directorate of Health and Integration, Lazio Region, 00145 Rome, Italy; abarca@regione.lazio.it; 15Scientific Direction, National Institute for Infectious Diseases Lazzaro Spallanzani IRCCS, 00149 Rome, Italy; enrico.girardi@inmi.it

**Keywords:** mpox infection, mpox vaccination, risk awareness, vaccine acceptance, vaccine hesitancy, health-related quality of life, the short-form 36-item questionnaire

## Abstract

Background: we aim to investigate attitudes toward vaccination by analyzing empirical factors associated with vaccine acceptance in the Lazio region mpox vaccination (MpoxVax) campaign in Italy. Methods: all subjects who accessed MpoxVax and signed the informed consent were prospectively enrolled in the MPOX-VAC Study and were asked to fill out an anonymous survey. Two endpoints were selected: ‘delayed acceptance’ and ‘early acceptance’, defined as access for vaccination >60 and ≤30 days from the vaccination campaign starting (VCS), respectively. Results: over the study period, 1717 individuals underwent vaccination: 129 (7%) > 60 [1588 (92.5%) ≤ 60] and 676 (60%) ≤ 30 days from VCS. A bisexual orientation, a lower education level and a worse perceived physical and mental health were associated with delayed access to vaccination. Being pre-exposure prophylaxis (PrEP) users and, marginally, HIV positive; having a high perceived risk for mpox infection; and reporting high-risk behaviors like the use of recreational drugs/chems, sex under the influence of drugs and/or alcohol and having a higher number of principal sexual partners, were associated with early access to vaccination. Conclusions: according to our data, risk awareness was a major determinant of early MpoxVax acceptance. Conversely, worse perceived health status and a low educational level were critical factors associated with delayed vaccination.

## 1. Introduction

### 1.1. Mpox Outbreak and Vaccination

On May 2022, a rapidly spreading mpox outbreak appeared, and it was declared a Public Health Emergency of International Concern by the World Health Organization (WHO), on 23 July 2022 [1,2]. Compared to all previous mpox epidemic events, the 2022–2023 outbreak presented a peculiar pattern in terms of transmission-associated factors, an unusually high frequency of interhuman transmission, and clinical presentation [3,4,5,6]. For this reason, vaccination against mpox (MpoxVax) was promptly recommended for individuals at high risk of exposure. Even in Italy, the third-generation modified, non-replicating live Ankara vaccine virus, produced by Bavarian Nordic MVA-BN, was authorized by the Italian Ministry of Health as mpox pre-exposure prophylaxis in adults at high risk of infection [7]. However, while vaccination is often cited as one of the most effective methods to control the spread of infectious diseases, some individuals are still hesitant to accept receiving vaccinations.

### 1.2. How to Define Vaccine Hesitancy and Vaccine Acceptance

Vaccine hesitancy, defined as delayed acceptance or refusal of vaccination despite the availability of vaccine services [8], was identified by the WHO as one of the top 10 threats to global health in 2019 [9], and it remains a widespread problem in the general population. More generally, vaccine acceptance is defined as the individual or group decision to accept or refuse, when presented with an opportunity to vaccinate, and it can be active (adherence by an informed public that perceives the benefit of and the need for a vaccine) or passive (compliance by a public that defers to recommendations and social pressure) [10,11]. It is well recognized that vaccination intention does not always correlate with actual behavior, and similarly, vaccine acceptance is not synonymous with vaccine uptake. To date, most studies focused on vaccine acceptance have assessed individuals’ intentions to receive the vaccine, rather than their explicit acceptance of the available vaccine itself [12,13,14].

### 1.3. Factors Associated with Vaccine Hesitancy

Hesitation towards vaccines is context-specific and influenced by factors such as convenience, confidence, risk perception, and ease of access to disease information and immunization services [8]. Moreover, sociodemographic factors such as age, gender, geographic area of residence, fear of adverse effects, or distrust of medical personnel and the health care system may influence vaccination decision making [12,15]. A number of studies have been published evaluating people’s willingness to accept MpoxVax, including both the general population [16] and high-risk individuals [8,17,18,19]; but, with the exception of a few recent reports focusing on MpoxVax uptake in a small number of subjects [20,21,22], data on its determinants are lacking.

### 1.4. Willingness to Accept and Receive MpoxVax

The number of vaccine doses administered in the United States was approximately 1.2 million, between 22 May 2022 and 31 January 2023, suggesting that only 23% of the at-risk population has been fully vaccinated [2]. As of 3 March, a total of 336,976 vaccine doses had been administered in 25 EU/EEA countries, of which 25,809 were in Italy [23]. It is likely that vaccination coverage varies widely among countries and states, depending on the degree of accessibility and awareness of vaccines, the number of vaccine providers, and the degree of trust and concern about the disease. In the Lazio region, vaccination coverage has been estimated at around 44% of potentially eligible people, and in order to increase uptake, tools need to be identified [24]. We hypothesize that factors such as demographics, ethnicity, educational level, HIV status, use of pre-exposure prophylaxis (PrEP), perceived quality of information received about MpoxVax, perceived risk of mpox compared to the general population, sexual orientation and conduct, and health-related quality of life (HRQoL) may influence the decision to get vaccinated. Therefore, we planned to measure the willingness to accept and receive MpoxVax by analyzing these associated empirical factors with a survey administered at the time of vaccination to participants in the MPOX-VAC study conducted during the MpoxVax campaign in the Lazio region of Italy.

## 2. Materials and Methods

### 2.1. Study Design, Study Population and Ethics

MPOX-VAC is an ongoing prospective observational and monocentric study conducted at the National Institute for Infectious Diseases “L. Spallanzani” IRCCS in Rome, Italy, enrolling high-risk persons who underwent MpoxVax, with the aims of monitoring safety, efficacy, immunogenicity, and acceptability of the MVA-BN vaccine. The study protocol (MPOX-VAC Study, version 1.0, 23 August 2022), was approved by the Ethical Committee of the Institute (approval number:41-z 2022), and all the included subjects signed an informed consent (version 1.0, 23 August 2022) for study participation and processing of personal data. Adult individuals (18 years or older) who met the criteria for priority access to MpoxVax, according to current Ministerial guidelines [3], and had access to MpoxVax at the National Institute of Infectious Diseases ‘L. Spallanzani’ IRCCS, the only regional hub in Lazio, were considered eligible for the study. The target population of the vaccination campaign, according to the Ministry of Health indications, was defined as gay, bisexual, or other MSM (men who have sex with men), reporting multiple sexual partners; participation in group sex events; sexual encounters in clubs/cruises/saunas; recent sexually transmitted infections; or sexual acts associated with the use of chemical drugs. Subjects who were unable or refused to sign the informed consent, those excluded from vaccination due to clinical contraindications, ongoing acute illness, or previous mpox infection, and those lacking basic knowledge of the Italian language, were excluded from the study. Dedicated internet pages on institutional and association websites were used to disseminate information on the vaccination campaign to the target population. Herein, we report the results of a cross-sectional analysis including all the individuals referred to the MpoxVax campaign in the Lazio region, from 8 August 2022 (vaccination campaign start) to 13 January 2023, and enrolled in the MPOX-VAC study. Individuals were asked to fill out an anonymous survey of 17 multiple-choice questions on demographics, perceived risk for mpox infection, sexual behavior, vaccination attitude, and perceived health status (Short Form Health Survey 36 questionnaire; SF-36). All the information (clinical, demographic, and behavioral data) was collected in an Electronic Case Report Form (eCRF), participants were identified by numeric codes only, and password protected.

### 2.2. Vaccine Adherence Questionnaire

In this study, the instrument used to conduct the behavioral survey was identified and constructed by a team of neuropsychologists, infectious diseases, and epidemiology specialists. The questionnaire, consisting of 17 items, was divided into two sections: (i) the demographic profile section, exploring demographic and biographical information such as sexual orientation, age, ethnicity, education, and work activity; in addition, individuals were asked to indicate their source of information for MpoxVax and the perceived quality of the information received, to indicate their motivation for vaccination (whether voluntary or under medical indication). And (ii) the sexual conduct section, inquiring about the number of partners in the last month, use of contraceptives, number of sexual intercourses with HIV-infected individuals, use of substances and alcohol, number of sexual intercourses under the use of substances and alcohol, and, finally, exploring the perceived risk of acquiring mpox compared to the general population based on sexual behaviors.

### 2.3. SF-36 Questionnaire

HRQoL and self-perception of health status [25] were assessed through the administration of the validated Italian version of the SF-36 questionnaire [26], a 36-item self-administered questionnaire with a high degree of reliability [27,28]. Eight health domains can be obtained from the SF-36: physical role functioning (PF, 10 items), role limitations–physical (RP, 4 items), bodily pain (BP, 2 items), general health perceptions (GH, 5 items) pertaining to physical health (PH), besides vitality (VT, 4 items), social role functioning (SF, 2 items), emotional role functioning (RE, 3 items), and mental health (MH, 5 items) pertaining instead to mental health (MH). The 8 domains contribute to two different scores: a mental health component summary (MCS) and a physical health component summary (PCS). Individuals can rate their responses on a three- or six-point scale and the summed scores of those responses are then coded and transformed into a scale from 0 (worst health) to 100 (best health) [29].

### 2.4. Statistical Analyses

In order to guarantee a representative sample size of the target population of subjects eligible for MpoxVax in Lazio region (N = 5400 MSM) [24], on the key exposures of interest with no more of 2% of margin of error, the required minimal sample size was of 1663 subjects. Descriptive characteristics were provided using medians and interquartile ranges (IQR) for continuous variables, and frequencies and percentages for categorical ones. Chi-square and Wilcoxon rank-sum tests were used to compare participants’ characteristics and survey responses in the two groups (vaccination ≤60 versus >60 days from the vaccination campaign start). Two endpoints were established: a) ‘delayed acceptance’ of MpoxVax (defined as access to vaccination more than 60 days from the campaign starting), and b) ‘early acceptance’ of MpoxVax (defined as access to vaccination less than 30 days from the campaign starting). These definitions were designed specifically for our study population based on the observation of the proceeding of new cases of mpox infection in Italy, and, therefore, the perceptions of the epidemics in high-risk subjects (N = 289 mpox cases in the first month of vaccination and N = 67 mpox cases in the second month after the start of the vaccination campaign in Italy). Logistic regression models were used to assess the association between demographic/behavioral factors and the two endpoints. The following factors were investigated as potential predictors of the two selected endpoints: age, sexual orientation, HIV status, PrEP use, ethnicity, perceived risk of mpox, perceived quality of information on vaccination, educational level, number of sexual partners, use of drugs/chemsex and alcohol, and MCS and PCS scores of the SF-36. Potential confounders, and adjustment sets, for each of the exposures of interest were identified according to the assumptions shown in the directed acyclic graph (DAG) in Appendix A. Statistical analyses were performed using Stata (v.14). All *p* values presented are two-sided, with *p* < 0.05 indicating statistical significance.

## 3. Results

### 3.1. Descriptive Analysis

Overall, 1717 participants answered the questionnaires and underwent vaccination. As shown in Figure 1, the vaccination campaign started on 8 August 2022, and was characterized by an initial extensive adherence, especially in the first 2 months of the campaign, and a subsequent progressive decrease over time.

General characteristics of the study population and comparisons between the two groups are shown in Table 1. Among 1717 participants, 129 individuals (7%) had delayed access to vaccination (>60 days from the vaccination campaign start) and 1588 (92.5%) had early access (<30 days). In particular, in the first group of 129 individuals, the median age was 38 (IQR 31–46), with most respondents older than 45 years (87, 67.4%), and 105 (81.4%) were Caucasian, while in the second group of 1588 subjects, the median age was 39 (33–46), 1092 (68. 8%) were older than 45 years, and 1354 (85.2%) were Caucasian. Participants were mainly HIV negative (63.6% versus 72%, *p* = 0.074); among these 191, 11.1% were on PrEP (6.2% versus 11.5%, *p* = 0.048). With regards to sexual orientation and conduct, participants were mainly homosexual (84.5% versus 93.3%, *p* < 0.001), and the proportions of respondents reporting a higher perceived risk of contracting mpox disease compared to the general population (57.3% versus 63.5%, *p* = 0.002), the use of recreational drugs during the last month (17.8% versus 17.3%, *p* = 0.933), and sexual intercourse with alcohol or drugs in the last month (18.6% versus 20.3%, *p* = 0.875) were similar between the two groups.

### 3.2. Analysis of ‘Delayed Acceptance’

By fitting separate multivariate logistic regression models for each of the exposures of interest on a delayed acceptance endpoint, we observed that bisexual orientation (versus homosexual, adjusted odds ratio (AOR) 3.22; 95% confidence interval (CI) 1.77–5.84), lower education level (versus high school/university, AOR 3.65; 95%CI 1.83–7.28), and reporting a worse perceived physical (per 10 points lower of SF-36 PCS, AOR 1.16; 95%CI 1.02–1.32) and mental health status (per 10 points lower of SF-36 MCS, AOR 1.13; 95%CI 1.02–1.23) were associated with delayed vaccination. On the contrary, participants with a high perceived risk for mpox infection compared to the general population, showed a lower risk for delayed access to the vaccine (AOR 0.61; 95%CI 0.41–0.91) (Table 2).

### 3.3. Analysis of ‘Early Acceptance’

Using the same approach for the early acceptance endpoint, being PrEP users, and marginally, HIV positive (versus HIV negative not on PrEP, AOR 1.97; 95%CI 1.37–2.82 and AOR 1.24; 95%CI 0.99–1.57, respectively), a bisexual orientation (versus homosexual, AOR 0.29; 95%CI 0.18–0.47), having a high perceived risk of mpox infection (AOR 1.43; 95%CI 1.13–1.82) and reporting high-risk behaviors like the use of recreational drugs (AOR 1.49, 95%CI 1.11–2.00), sex under the influence of drugs and/or alcohol (AOR 1.78; 95%CI 1.36–2.34) and a higher number of principal sexual partners (per one more, AOR 1.07; 95%CI 1.03–1.11) were associated with early vaccination, along with receiving poor or no information on MpoxVax (AOR 2.91; 95%CI 1.53–5.55) (Table 3).

## 4. Discussion

### 4.1. Risk Awareness as a Key Determinant of Vaccine Uptake

Our data show that, in our population, individuals with a high degree of risk awareness for mpox infection and reporting high-risk behaviors, such as PrEP users and people living with HIV, have access to vaccines in the first 30 days of the vaccination campaign. These results are in accordance with the prioritization criteria of the Italian vaccination campaign. Conversely, a bisexual orientation, a lower education level, and reporting a worse perceived physical and mental health status compared to the general population were critical factors associated with delayed vaccination access. We therefore see how a number of local, racial, and cultural factors, along with several other aspects, including misinformation, can influence people’s perceptions of vaccination acceptance, as it was clearly observed in the recent COVID-19 pandemic and related vaccination [30]. In particular, people’s psychological state was found to have an influence on their attitude toward the COVID-19 vaccine: subjects reporting a generally good psychological state and having previously received a vaccine were found to show a more positive intention to be vaccinated [31]; in contrast, a higher external locus of control over health, intended as the idea that one’s health highly depends on chance and external factors more than personal control, was directly associated with hesitant or negative vaccination intentions [31]. Perception of the risk, which also impacts vaccination attitude, is strongly influenced by communication from media, government, and scientific institutes and by the trust/mistrust in these sources, especially when people believe they have the capacity to have control over these risky behaviors [32]. Therefore, important information-related factors that determine vaccination intentions are the value placed on science and the attitudes to perceive media bias and misinformation [33].

### 4.2. Willingness to Accept MpoxVax among High-Risk Individuals

In a recent Dutch study conducted among unvaccinated high-risk individuals, willingness to accept MpoxVax was observed in 1859 subjects (938 eligible for primary preventive vaccination and 918 not eligible). The proportion of those willing to accept vaccination was 81.5%, similar between eligible (85%) and not (78%) [17]. Similar results from the current literature have indicated that in Europe, there appears to be a high prevalence of acceptance of the MpoxVax (70%) compared to Asia (50%), probably due to the higher incidence of mpox being associated with a greater perception of risk [34]. Depending on the risk of contracting the disease, therefore, vaccine acceptance was greater in the LGBTI population (84%) compared to the general population (43%) [34]. All our study participants asked for vaccination spontaneously, particularly PrEP users and people living with HIV, because they felt more at risk than the general population. This finding is partially consistent with a French survey that reported that of 402 PrEP users, 369 (87.0%) have been vaccinated against mpox, most of whom had sought vaccination spontaneously during the summer of 2022. Interestingly, half of the PrEP users who refused vaccination did not feel themselves at risk, probably because, as also described in another French survey, MSM on PrEP reported having few sexual partners [18,20].

### 4.3. The Role of Other Determinants as Psychosocial Factors and Perception of Health Status

Thus, among the most frequently reported beliefs for MpoxVax acceptance, we found risk awareness and motivation to protect oneself from infection. Public health communication messages should include more factual information about the risk of exposure, transmission routes, symptoms, and side effects of the vaccine through awareness-raising campaigns on institutional and non-institutional websites and social platforms. This should encourage a person who has a high risk of exposure to feel themself at risk and to evaluate mpox as potentially serious and the vaccine as beneficial. On the contrary, a lower level of education was a critical factor for delaying vaccination; a similar result was found in a recent study in which the odds of being neutral (as opposed to being willing) towards the MpoxVax were higher for those with a lower level of education [17]. Bisexual orientation also emerges as a critical factor for vaccination delay; similarly, in a study comparing the socio-demographic characteristics of eligible participants according to vaccination status, statistically significant differences were found according to gender identity and sexual identity. Out of a sample of 331 participants, not being vaccinated was more common, along with other characteristics, among bisexuals (31.4% unvaccinated versus 9.4% vaccinated, *p* < 0.001) [22]. Not surprisingly, the perception of a worse state of physical and mental health in those who access late vaccination could explain the lack of motivation to vaccinate earlier and the increased fear of the side effects of the vaccine on their own condition of health.

### 4.4. Strengths

The strengths of this study were mainly represented by the large sample size and the timing of the survey, in terms of timeliness of data recording, with data collection beginning close to the start of the vaccination campaign, and a large investigation period (six months of data collection), together with the evaluation of a wide range of epidemiological determinants.

### 4.5. Limitations

The limitations of our study were, first of all, a probable self-choice bias of our sample, consisting essentially of individuals who voluntarily underwent vaccination, and a cross-sectional study, so no causality could be established. Also, our sample did not include women. Although MSM constitute the majority of current mpox cases in the United States and European countries, all are at risk regardless of their sexual identity. Another limit is the fact that we used a single-center analysis, which may impact the generalizability of the results, even though it was the only regional reference center, and the characteristics of the study participants conform to the vaccination target population. In addition, the lack of a comparison sample composed of unvaccinated subjects upon which to associate our data represents another important limitation of the study, so further research needs to be conducted in these terms.

## 5. Conclusions

Vaccine hesitancy and acceptance are key determinants of vaccination coverage that should be assessed and consequently addressed with evidence, education, and promotion as part of disease prevention campaigns, including mpox. Therefore, even in the case of mpox, it seems critical to emphasize the need for immunization as an essential public health intervention to contain infection transmission and disease development. In our study, risk awareness was confirmed as a major determinant of vaccination acceptance, along with the need to receive good-quality information about the disease and vaccination. Moreover, our results may be considered useful in directing future vaccination campaigns in order to protect individuals who belong to the categories with a lower physical/mental health and socio-cultural status, as these groups also in our study showed a higher degree of hesitancy toward vaccination, despite being potentially at greater risk of complications and worse outcomes—particularly the first group. The success of this process relies on the ability of institutions and media to be able to bring clear and shared information. Healthcare professionals, who have earned the trust of their patients with a direct relationship, as for PLWH and PrEP users in our study, can also aid in the vaccination drive and in the development of a critical capacity of the subjects in interpreting and assessing risks in both absolute terms and in relation to their status.

In conclusion, the findings of our study could be a useful tool, at a public health level, for identifying strategies to encourage vaccine acceptance and address vaccine uncertainty or increase uptake.

## Figures and Tables

**Figure 1 vaccines-11-01761-f001:**
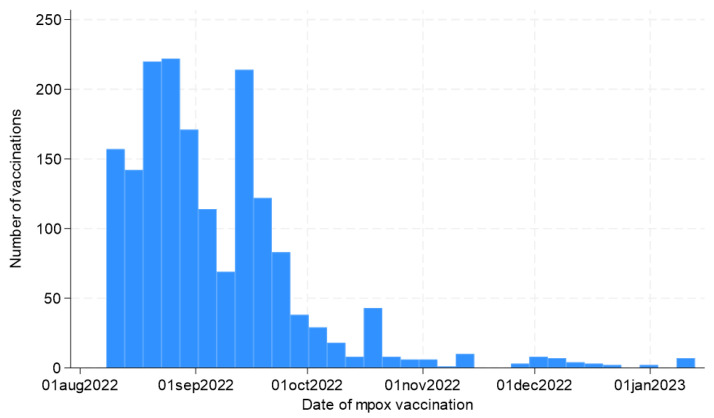
Number of accesses for mpox vaccination and enrollments in the MPOX-VAC study over the study period (8 August 2022–13 January 2023).

**Table 1 vaccines-11-01761-t001:** Characteristics and survey responses of the study population and comparison between the two groups.

	>60 Days from Start of Campaign	≤60 Days from Start of Campaign	*p* Values	Total
	N = 129 (7.5%)	N = 1588 (92.5%)		N = 1717 (100%)
**Age, median (IQR)**	38 (31–46)	39 (33–46)	0.233	39 (33–46)
Age > 45 years, *n* (%)	87 (67.4)	1092 (68.8)	0.755	1179 (68.7)
**Sexual orientation, n (%)**			<0.001	
Homosexual	109 (84.5)	1481 (93.3)		1590 (92.6)
Bisexual	15 (11.6)	64 (4.0)		79 (4.6)
Transgender	1 (0.8)	14 (0.9)		15 (0.9)
Other MSM	3 (2.3)	25 (1.6)		28 (1.6)
Unknown	1 (0.8)	4 (0.2)		5 (0.3)
**HIV status, n (%)**			0.074	
Negative	82 (63.6)	1144 (72.0)		1226 (71.5)
Positive	46 (35.7)	443 (27.9)		489 (28.5)
Missing	1 (0.8)	1 (0.1)		2 (0.1)
On PrEP, n (%)	8 (6.2)	183 (11.5)	0.048	191 (11.1)
**Ethnicity, n (%)**			0.05	
Caucasian	105 (81.4)	1354 (85.2)		1459 (85.0)
Black	0 (0.0)	13 (0.8)		13 (0.7)
Asian	3 (2.3)	11 (0.6)		14 (0.8)
Other	18 (14)	142 (8.9)		160 (9.3)
Missing	3 (2.3)	68 (4.2)		71 (4.1)
**Education, years, n (%)**			0.002	
<5	3 (2.3)	7 (0.4)		10 (0.5)
8	9 (7)	35 (2.2)		44 (2.5)
13	36 (28)	472 (29.7)		508 (29.5)
18	49 (38)	623 (39.2)		672 (39.1)
>18	29 (22.4)	411 (25.8)		440 (25.6)
Missing	3 (2.3)	40 (2.5)		43 (2.5)
**Voluntary choice of mpox vaccination, n (%)**			0.498	
Yes personal voluntary choice	122 (95.3)	1516 (96.6)		1638 (96.5)
Under medical indication	5 (3.9)	36 (2.3)		41 (2.4)
Other/Unknown	1 (0.8)	17 (1.1)		18 (1.1)
**Quality of info on vaccine, n (%)**			0.001	
Optimal	112 (86.2)	1312 (82.6)		1424 (83)
Good	6 (4.6)	67 (4.2)		73 (4.2)
Poor	0 (0.0)	20 (1.2)		20 (1.1)
None	1 (0.7)	41 (2.5)		42 (2.4)
missing	8 (6.2)	147 (9.2)		155 (9.0)
**Perception of Risk, n(%)**			0.002	
Equal	46 (35.6)	372 (23.4)		418 (24.3)
Lower	8 (6.2)	97 (6.0)		105 (6.1)
Higher	74 (57.3)	1009 (63.5)		1083 (63.0)
Missing	1 (0.7)	110 (7)		111 (6.4)
**Number of main sexual partners, median (IQR)**	0 (0–0)	2 (1–4)	0.098	2 (1–4)
**Use of recreational drugs/chemsex last month, n (%)**			0.933	
Yes	23 (17.8)	275 (17.3)		298 (17.3)
No	99 (76.7)	1242 (78.2)		1341 (78.1)
Do not know	1 (0.7)	6 (0.3)		7 (0.4)
Rather not answer	2 (1.5)	28 (1.7)		30 (1.7)
Missing	4 (3.1)	37 (2.3)		41 (2.4)
**Sexual intercourse with alcohol or chems in last month, n (%)**			0.875	
No	97 (75.1)	1172 (73.8)		1269 (73.9)
Yes	24 (18.6)	323 (20.3)		347 (20.2)
Unknown	8 (6.2)	93 (5.9)		101 (5.9)
**SF-36 questionnaire scales, median (IQR)**				
SF36 Physical Functioning	100 (95–100)	100 (100–100)	0.221	100 (100–100)
SF36 Role—Physical	100 (100–100)	100 (100–100)	0.16	100 (100–100)
SF36 Bodily Pain	100 (74–100)	100 (80–100)	0.91	100 (80–100)
SF36 General Health	72 (56–82)	76 (61–85)	0.128	75 (61–85)
SF36 Vitality	60 (55–75)	65 (55–75)	0.322	65 (55–75)
SF36 Social Functioning	87 (62–100)	87 (62–100)	0.027	87 (62–100)
SF36 Role—Emotional	100 (33–100)	100 (66–100)	0.003	100 (66–100)
SF36 Mental Health	68 (56–84)	72 (60–84)	0.098	72 (60–84)

**Table 2 vaccines-11-01761-t002:** Odds ratio (OR) and adjusted OR (AOR) of being vaccinated after 60 days from the start of the mpox vaccination campaign in the Lazio region (delayed acceptance), using logistic regression analysis.

	OR	95%CI	*p* Values	AOR *	95%CI	*p* Values
**Age, per 10 years more**	0.9	0.75–1.10	0.31	0.92	0.76–1.11	0.373
Sexual orientation						
Homosexual	1			1		
Bisexual	3.18	1.76–5.77	<0.001	3.22	1.77–5.84	<0.001
Transgender	0.97	0.13–7.45	0.977	1	0.13–7.67	0.999
Other	1.63	0.48–5.49	0.43	1.67	0.49–5.62	0.409
**HIV/PrEP status**						
HIV−/No PrEP	1			1		
HIV+	1.35	0.92–1.99	0.125	1.34	0.9–2.00	0.151
HIV−/On PrEP	0.58	0.27–1.21	0.146	0.62	0.29–1.33	0.22
**Ethnicity, Caucasian (vs. non-Caucasian)**	0.61	0.37–1.01	0.053	0.63	0.38–1.03	0.065
**Education, Middle/Elementary (vs. High school/University)**	3.77	1.93–7.37	<0.001	3.65	1.83–7.28	<0.001
**Quality of information on mpox vaccination**						
Optimal Good	1					
Poor None	0.19	0.03–1.39	0.103	0.16	0.02–1.20	0.075
Unknown	0.79	0.41–1.54	0.488	0.68	0.34–1.37	0.282
**Self-Perception of risk compared to general population**						
Equal	1			1		
Lower	0.67	0.30–1.46	0.311	0.66	0.30–1.47	0.312
Higher	0.59	0.40–0.87	0.008	0.61	0.41–0.91	0.015
Unknown	0.07	0.01–0.54	0.01	0.07	0.01–0.55	0.011
**N. main partners, per 1 more**	0.93	0.86–1.01	0.084	0.94	0.86–1.01	0.103
**Use of Drugs/Chems, yes (vs. no)**	1.05	0.65–1.68	0.842	1.14	0.70–1.85	0.596
**Sex under the influence of drugs or alcohol**						
No	1			1		
Yes	0.9	0.56–1.43	0.648	0.94	0.59–1.51	0.799
Unknown	1.04	0.49–2.20	0.92	0.88	0.34–2.27	0.796
**SF-36 PCS, per 10 pt decrease**	1.14	1.01–1.30	0.039	1.16	1.02–1.32	0.028
**SF-36 MCS, per 10 pt decrease**	1.13	1.03–1.24	0.008	1.12	1.03–1.23	0.011

* Set of Adjustments: Age was adjusted for ethnicity; sexual orientation, ethnicity, MCS, and PCS were adjusted for age; HIV/PrEP status was adjusted for use of drugs/alcohol for sex, age, ethnicity, principal partner, perception of risk, and education; education was adjusted for ethnicity and age; perception of risk was adjusted for age, drugs/alcohol, sexual orientation, and principal partner; N. of principal partners was adjusted for age and sexual orientation; use of drugs/chems and sex under drugs/alcohol was adjusted for age sexual orientation and MCS; quality of information on mpox vaccination was adjusted for age, ethnicity, and education. Abbreviations: mpox, formerly named monkeypox; MCS, mental component summary; n, number of participants; PrEP, pre-exposure prophylaxis; PCS, physical component summary; vs., versus.

**Table 3 vaccines-11-01761-t003:** Odds ratio (OR) and adjusted OR (AOR) of being vaccinated in the first 30 days of the mpox vaccination campaign in the Latium region (early acceptance), using logistic regression analysis.

	OR	95%CI	*p*	AOR *	95%CI	*p*
**Age, per 10 years more**	1.05	0.95–1.17	0.336	1.05	0.95–1.16	0.361
**Sexual orientation**						
Homosexual	1			1		
Bisexual	0.29	0.18–0.48	<0.001	0.29	0.18–0.47	<0.001
Transgender	0.53	0.19–1.46	0.217	0.52	0.19–1.43	0.203
Other	0.52	0.25–1.10	0.088	0.51	0.24–1.09	0.081
**HIV/PrEP status**						
HIV−/No PrEP	1			1		
HIV+	1.23	0.98–1.53	0.07	1.24	0.99–1.57	0.064
HIV−/On PrEP	2.11	1.49–2.99	<0.001	1.97	1.37–2.82	<0.001
**Ethnicity, Caucasian (vs. non-Caucasian)**	1.27	0.93–1.72	0.132	1.25	0.92–1.71	0.151
**Education, High school/University (vs. Middle/Elemenary)**	1.75	1.01–3.01	0.044	1.7	0.98–2.94	0.06
**Quality of information on mpox vaccination**						
Optimal Good	1			1		
Poor None	2.79	1.48–5.29	0.002	2.91	1.53–5.55	0.001
Unknown	1.01	0.72–1.41	0.947	1.07	0.76–1.5	0.706
**Self-Perception of risk compared to general population**						
Equal	1			1		
Lower	0.91	0.59–1.39	0.661	0.93	0.60–1.44	0.749
Higher	1.52	1.21–1.91	<0.001	1.43	1.13–1.82	0.003
Unknown	7.64	4.07–14.33	<0.001	6.9	3.65–13.02	<0.001
**N. main partners, per 1 more**	1.07	1.03–1.11	0.001	1.07	1.03–1.11	0.001
**Use of Drugs/Chems, yes (vs. no)**	1.54	1.18–2.01	0.002	1.49	1.11–2.00	0.007
Sex under the influence of drugs or alcohol						
No	1			1		
Yes	1.67	1.3–2.15	<0.001	1.78	1.36–2.34	<0.001
Unknown	2.04	1.3–3.22	0.002	2.04	1.18–3.53	0.01
**SF36 PCS, per 10 pt decrease**	1.10	1.01–1.20	0.033	1.10	1.01–1.20	0.026
**SF36 MCS, per 10 pt decrease**	1.00	0.94–1.05	0.908	0.99	0.94–1.05	0.839

*** Set of Adjustments: Age was adjusted for ethnicity; sexual orientation, ethnicity, MCS, and PCS were adjusted for age; HIV/PrEP status was adjusted for use of drugs/alcohol for sex, age, ethnicity, principal partner, perception of risk, and education; education was adjusted for ethnicity and age; perception of risk was adjusted for age, drugs/alcohol, sexual orientation, and principal partner; N. of principal partners were adjusted for age and sexual orientation; use of drugs/chems and sex under drugs/alcohol were adjusted for age, sexual orientation and MCS; quality of information on mpox vaccination was adjusted for age, education, and ethnicity. Abbreviations: Mpox, formerly named monkeypox; MCS, mental component summary; n, number of participants; PrEP, pre-exposure prophylaxis; PCS, physical component summary; vs., versus.

## Data Availability

The raw data generated and/or analyzed within the present study are available in our institutional repository (rawdata.inmi.it), subject to registration. In the event of a malfunction of the application, the request can be sent directly by e-mail to the Library (biblioteca@inmi.it). No charge for granting access to data is required.

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
