# Peer review of "Risk Awareness as a Key Determinant of Early Vaccine Uptake in the Mpox Vaccination Campaign in an Italian Region: A Cross-Sectional Analysis"

_vaccines, 2023, doi:10.3390/vaccines11121761_

Round 1

Reviewer 1 Report

Comments and Suggestions for Authors

Thank you very much for performing this research and sharing your data with the scientific community. 

I believe this manuscript, if accepted for publication, will be well-read and cited.

Good luck

1. The main aim of the study is to define the vaccine hesitancy and vaccine uptake habits of Mpox vaccine recipients. 

2. This is a original topic and there still is scarcity of information regarding Mpox vaccines. As it's been known Mpox is a sexually transmitted infection prevalent among immunocompromised people including PLWH and people with risk taking behaviors. There is a certain information gap regarding vaccine uptake in this population. And as you know future vaccine candidates aiming other sexually transmitted infections are now underdevelopment (please see the recent manuscript https://www.thelancet.com/journals/lanepe/article/PIIS2666-7762(23)00157-6/fulltext) 

the findings of the Italian study is therefore has an pivotal importance.   3. It defines the cohort of people who have vaccine hesitancy and compares their attitudes with people who accepts vaccination as early as it became available. 4. The references are appropriate.

Author Response

RESPONSE TO REVIEWER 1 COMMENTS

Dear Reviewer 1, thank you very much for taking the time to review this manuscript. Please find the detailed responses below and the corresponding revisions highlighted in the re-submitted file.

POINT-BY-POINT RESPONSES

Thank you very much for performing this research and sharing your data with the scientific community. 

I believe this manuscript, if accepted for publication, will be well-read and cited.

Good luck

  1. The main aim of the study is to define the vaccine hesitancy and vaccine uptake habits of Mpox vaccine recipients. 
  2. This is a original topic and there still is scarcity of information regarding Mpox vaccines. As it's been known Mpox is a sexually transmitted infection prevalent among immunocompromised people including PLWH and people with risk taking behaviors. There is a certain information gap regarding vaccine uptake in this population. And as you know future vaccine candidates aiming other sexually transmitted infections are now underdevelopment (please see the recent manuscript Https://www.thelancet.com/journals/lanepe/article/PIIS2666-7762(23)00157-6/fulltext) the findings of the Italian study is therefore has an pivotal importance.  
  3. It defines the cohort of people who have vaccine hesitancy and compares their attitudes with people who accepts vaccination as early as it became available.
  4. The references are appropriate.

Response. Dear reviewer, we really appreciate your comments and we agree with you on the current lack of information on vaccine uptake in this population. With the aim of updating our knowledge, we have investigated and added the study you recommended among the bibliographical references (number 20). We have also added in the introduction a new paragraph on mpox vaccine update according to WHO surveillance data (lines 90-92).

Reviewer 2 Report

Comments and Suggestions for Authors

The article presented for review is interesting and written in clear language. It has the correct structure. The tests used are correct. Precisely because of the nature of the tests, the tables showing the results are quite large and take up a lot of space in the text.

I have three comments:
1. The size of the group >60 days from start of campaign amounting to N=129, which is 7.5% of the total study group, seems a bit too small, which may affect the results of comparative analyses;
2. Such a small group size may also affect the results of regression analysis in this group.

3. Recently, there have been quite a few interesting papers related to various types (including many on psychosocial determinants) of COVID-19-related vaccination decision-making. Might it be worthwhile to expand the discussion of results by analyzing a few papers on this topic?

Author Response

RESPONSE TO REVIEWER 2 COMMENTS

Dear Reviewer 2, thank you very much for taking the time to review this manuscript. Please find the detailed responses below and the corresponding revisions highlighted in the re-submitted file.

POINT-BY-POINT RESPONSES

The article presented for review is interesting and written in clear language. It has the correct structure. The tests used are correct. Precisely because of the nature of the tests, the tables showing the results are quite large and take up a lot of space in the text.

I have three comments:
1. The size of the group >60 days from start of campaign amounting to N=129, which is 7.5% of the total study group, seems a bit too small, which may affect the results of comparative analyses;

  1. Such a small group size may also affect the results of regression analysis in this group.

Response. Thank you for pointing this out; the following answer refers to both of your comments.

Sample size calculation for logistic regression with this number of different predictors is complex, but there are general “rule of thumb” that are the minimum to conduct a proper analysis also for logistic regression, which are respected in our analysis: having at least 100 events (and also for the late vaccination endpoint the number of events are >100) and having at least 10 events per variable (Pedruzzi et al. 1996). According to the cited work of Pedruzzi et al. 1996, the following guideline for a minimum number of cases to include in your study can be suggested: N = 10 k / p (where N is the target sample size; k is the number of covariates in the model; and p the smallest of the proportions in the study population). Specifically, in our analysis, the maximum number of covariates reached is 6, and the smallest proportion is 7.5%, so 800 (6x10/0.075) patients are needed. So our study has a sample size that is more than twice this minimum requested number.

  1. Recently, there have been quite a few interesting papers related to various types (including many on psychosocial determinants) of COVID-19-related vaccination decision-making. Might it be worthwhile to expand the discussion of results by analyzing a few papers on this topic?

Response. As kindly suggested, the discussion was expanded including articles on the psychosocial determinants of vaccination decision-making related to COVID-19, in order to emphasize this point (lines 258-269).

Reviewer 3 Report

Comments and Suggestions for Authors

This study investigates attitudes toward vaccination, by analyzing factors associated with vaccine acceptance in the Lazio region mpox vaccination campaign in Italy. The manuscript is well-written, but there are few issues listed below.  I recommend publication only if these questions are totally clarified. 

1.  It is recommended to provide an overview of the global and national acceptance, hesitancy, and uptake rates of mpox vaccination in the introduction. Additionally, summarizing the gaps in knowledge identified after conducting the literature review would be beneficial.

2. In line 73, the authors mentioned," To date, most studies focused on vaccine acceptance..." Please cite more papers after this sentence, such as DOI: 10.3390/vaccines10091453

3. To enhance readability, it would be beneficial to divide your introduction into 3-4 paragraphs. The current version can be challenging to comprehend.

4. Please add a sample size planning.

5. How did the authors define "delayed acceptance of vaccination" and "early acceptance of vaccination".  Is there any evidence or study to support such a definition?

6. Please add the software that you used in the statistical analyses.

7. The reviewer reads the discussion while experiencing fatigue. It is recommended to avoid summarizing the discussion in a single paragraph and instead separate it into 4-5 paragraphs.

  •  

Comments on the Quality of English Language

NA

Author Response

RESPONSE TO REVIEWER 3 COMMENTS

Dear Reviewer 3, thank you very much for taking the time to review this manuscript. Please find the detailed responses below and the corresponding revisions highlighted in the re-submitted file.

POINT-BY-POINT RESPONSES

This study investigates attitudes toward vaccination, by analyzing factors associated with vaccine acceptance in the Lazio region mpox vaccination campaign in Italy. The manuscript is well-written, but there are few issues listed below.  I recommend publication only if these questions are totally clarified. 

  1. It is recommended to provide an overview of the global and national acceptance, hesitancy, and uptake rates of mpox vaccination in the introduction. Additionally, summarizing the gaps in knowledge identified after conducting the literature review would be beneficial.

Response. Agree. We have, accordingly, added several data on mpox vaccine uptake (lines 90-96) and a discussion of knowledge gaps can be found in sections 1.3 and 1.4. We hope that this is sufficient to respond to your comments.

  1. In line 73, the authors mentioned," To date, most studies focused on vaccine acceptance..." Please cite more papers after this sentence, such as DOI: 10.3390/vaccines10091453

Response. More papers have been cited, including the one you mentioned (line 78; references numbers 13, 14, 15).

  1. To enhance readability, it would be beneficial to divide your introduction into 3-4 paragraphs. The current version can be challenging to comprehend.

Response. As kindly suggested by the reviewer, we have, accordingly, divided the introduction into the following 4 paragraphs:

1.1 Mpox outbreak and vaccination (line 53)

1.2 How to define vaccine hesitancy and vaccine acceptance (line 66)

1.3 Factors associated with vaccine hesitancy (line 79)

1.4 Willingness to accept and receive MpoxVax (line 89)

  1. Please add a sample size planning.

Response. As already written in response to Reviewer 2, we included 1717 users in our study, that are an adequate sample size according to the proportions and the models used: “According to the work of Pedruzzi et al. 1996, the following guideline for a minimum number of cases to include in your study can be suggested: N = 10 k / p (where N is the target sample size; k is the number of covariates in the model; and p the smallest of the proportions in the study population). Specifically, in our analysis, the maximum number of covariates reached is 6, and the smallest proportion is 7.5%, so 800 (6x10/0.075) patients are needed. So our study has a sample size that is more than twice this minimum requested number.”

Moreover, we calculated the sample size needed in the study based on the number needed to reach in the survey for being representative of the target population. According to the estimates of our group (Vairo et al IJID 2022), the total target population of MSM eligible for mpox vaccination in Lazio was 5400 subjects (N), considering a confidence interval of 95% (z=1.96) and a margin of error (MOE) of no more of 2% on key exposures in our analysis and a sample proportion 0.5 (p), the calculated required sample size (n) for the study was 1663

(Sample size for known population: n = N*X / (X + N – 1), where X = Zα/22 *p*(1-p) / MOE2).

This information has been added to the manuscript, in the “Statistical analyses” paragraph (lines 166-169).

  1. How did the authors define "delayed acceptance of vaccination" and "early acceptance of vaccination".  Is there any evidence or study to support such a definition?

Response. Dear reviewer, thank you for pointing this out. There is currently no established evidence or study to support our arbitrary definitions of “delayed vaccination acceptance” (>60 days from start) and “early vaccination acceptance” (first 30 days). These definitions were designed specifically for our study population based on the observation of the proceeding of new cases of Mpox infection in Italy, and therefore the perceptions of the epidemics in high-risk subjects: according to surveillance data a high proportion of mpox cases have been reported in the first month after the start of the vaccination campaign in Italy (N= 282; 29.3% of total diagnosis till today) and then significantly reduced after the summer with 67 mpox cases reported in the second month of vaccination campaign (7% of total diagnosis till today).

This information has been added to the manuscript, in the “Statistical analyses” paragraph (line 176-180).

  1. Please add the software that you used in the statistical analyses.

Response. The software used for statistical analyses is Stata v.14, added in the methods section (line 188-189).

  1. The reviewer reads the discussion while experiencing fatigue. It is recommended to avoid summarizing the discussion in a single paragraph and instead separate it into 4-5 paragraphs.

Response. As kindly suggested by the reviewer, we have, accordingly, divided the discussion into the following 5 paragraphs:

4.1 Risk awareness as a key determinant of vaccine uptake (line 247)

4.2 Willingness to accept MpoxVax among high-risk individuals (line 270)

4.3 The role of other determinants as psychosocial factors and perception of health status (line 288-289)

4.4 Strengths (line 309)

4.5 Limitations (line 315).
